# Effect of Frozen Treatment on the Sensory and Functional Quality of Extruded Fresh Noodles Made from Whole Tartary Buckwheat

**DOI:** 10.3390/foods11243989

**Published:** 2022-12-09

**Authors:** Zicong Guo, Lijuan Wang, Ruge Cao, Ju Qiu

**Affiliations:** 1Key Laboratory of Precision Nutrition and Food Quality, Department of Nutrition and Health, China Agricultural University, No.17 Tsinghua East Road, Haidian District, Beijing 100083, China; 2College of Food Science and Engineering, Tianjin University of Science and Technology, Tianjin 300457, China

**Keywords:** Tartary buckwheat, frozen, extrude noodles, cooking quality, water distribution, starch retrogradation

## Abstract

Extruded noodles made from whole Tartary buckwheat are widely known as healthy staple foods, while the treatment of fresh noodles after extrusion is crucial. The difference in sensory and functional quality between frozen noodles (FTBN) and hot air-dried noodles (DTBN) was investigated in this study. The results showed a shorter optimum cooking time (FTBN of 7 min vs. DTBN of 17 min), higher hardness (8656.99 g vs. 5502.98 g), and less cooking loss (5.85% vs. 21.88%) of noodles treated by freezing rather than hot air drying, which corresponded to better sensory quality (an overall acceptance of 7.90 points vs. 5.20 points). These effects on FTBN were attributed to its higher ratio of bound water than DTBN based on the Low-Field Nuclear Magnetic Resonance results and more pores of internal structure in noodles based on the Scanning Electron Microscopy results. The uniform water distribution in FTBN promoted a higher recrystallization (relative crystallinity of FTBN 26.47% vs. DTBN 16.48%) and retrogradation (degree of retrogradation of FTBN 34.67% vs. DTBN 26.98%) of starch than DTBN, strengthening the stability of starch gel after noodle extrusion. FTBN also avoided the loss of flavonoids and retained better antioxidant capacity than DTBN. Therefore, frozen treatment is feasible to maintain the same quality as freshly extruded noodles made from whole Tartary buckwheat. It displays significant commercial potential for gluten-free noodle production to maximize the health benefit of the whole grain, as well as economic benefits since it also meets the sensory quality requirements of consumers.

## 1. Introduction

Tartary buckwheat (*Fagopyrum tataricum* (L). Gaertn) is used to consume as a whole grain, which is beneficial for human health because of its bioactive compounds, such as flavonoids [1]. The contents of rutin and quercetin, usually reaching up to 6.37~41.05 mg/g in total, are the most in Tartary buckwheat compared to all cereals and pseudocereal crops [2]. Noodles, as one of the most popular whole grain foods, are the best way to take advantage of the health benefit of Tartary buckwheat as a raw material of low glycemic index products [3]. However, the lack of gluten protein in Tartary buckwheat calls for extrusion treatment, which is different from the traditional calendering technique of wheat noodles to avoid too much cooking loss and breakage [4]. Extrusion is generally used in gluten-free noodle production to form starch gel, relying on starch gelatinization and adhesion [5]. Therefore, improving the sensory and functional quality of extruded Tartary buckwheat noodles has attracted considerable attention from research and the food industry.

The replacement of gluten protein with starch to form a skeletal network structure during extrusion is suitable for starch-based noodles made from whole Tartary buckwheat [5]. However, fresh extruded noodles are generally dehydrated and dried for storage, which leads to easy breakage after cooking due to problematic or inappropriate rehydration during boiling [6]. Hot air drying is the most common method used during industrial production for its efficiency and accurate temperature control compared to native air drying [6,7], as well as the lower cost and easier operation compared to other drying methods, such as microwaving, infrared, and vacuuming [8]. However, the sharp water loss during hot air drying has a side effect on the quality of starch noodles, leading to cracks on the surface of noodles and, thus, deterioration of the cooking quality, namely more cooking loss and a broken ratio [6]. Compared to dried noodles, frozen noodles with a shorter cooking time show a more similar quality to fresh noodles [3]. Especially for extruded noodles, the frozen treatment promotes starch retrogradation and thus strengthens the starch gel structure with less cooking loss and better sensory quality [9]. Given that not only the extrusion processing but also the subsequent treatment plays a key role in cooking quality, the frozen treatment seems a feasible way, but it lacks clear evidence of extruded noodles, not to mention the Tartary buckwheat noodles. Therefore, improving the quality of frozen extruded noodles can increase their commercial value in the frozen food market.

It was reported that quick-freezing at −20 °C can minimize the adverse effects of mechanical damage to wheat noodle quality caused by ice crystals [10], as well as the impact on starch retrogradation of oat roll [11]. There are still few studies on the changes in extruded noodle quality induced by freezing. The present study aimed to (1) investigate the differences in the sensory and functional quality of Tartary buckwheat noodles between hot air drying and frozen treatment after extrusion; (2) determine the key role in water distribution and starch retrogradation; (3) clarify the correlation of quality with the internal structure change to explain the mechanism of improvement by frozen treatment.

## 2. Materials and Methods

### 2.1. Materials

The Tartary buckwheat (No.2 Chuanqiao) was harvested in 2020, and the whole kernels were milled to flour by Huantai Biotechnology Co., Ltd., Chengdu, China. The flour composition (*w*/*w*) mainly consisted of 49.96% starch, 14.55% water, 3.48% total flavonoids, 12.34% fiber, 8.96% protein, and 1.49% ash. The chemicals (analytical grade) and flavonoid standards (rutin and quercetin) were purchased from Sigma-Aldrich (St. Louis, MO, USA).

### 2.2. Preparation of Tartary Buckwheat Noodles

The whole Tartary buckwheat noodles were prepared using a twin-screw extruder (SLG30-IV, Saibainuo Scientific Development Inc., Jinan, China). Following our previous research, the amount of added water was 40%, and the speed was 15 Hz. The temperature was controlled at 60–70 °C, 70–90 °C, 90–100 °C, and 70–60 °C, respectively. The fresh extruded noodles (TBN) were further processed by hot air drying at 60 °C for 3 h (DTBN) or by freezing at −20 °C for 48 h (FTBN).

### 2.3. Cooking Quality of Noodles

The cooking qualities of noodles, including optimum cooking time, water absorption, broken ratio, and cooking loss, were determined according to a reported method [12].

#### 2.3.1. Optimum Cooking Time

The noodles (10 g) were weighed and boiled in 200 mL of water. During the cooking process, one strand of noodle was squeezed every 30 s by a transparent glass plate to observe its hardcore. The optimum cooking time was determined when the hard core of the noodles disappeared.

#### 2.3.2. Water Absorption

The noodles were boiled until the optimum cooking time and were rinsed with cool water for 10 s immediately. Cooked noodles were collected and weighed accurately after removing surface water with filter paper at room temperature. The water absorption was calculated as follows:Water absorption (%) = (*m*_2_ − *m*_1_)/*m*_1_ × 100
where *m*_1_ is the weight of noodles before cooking (g), and *m*_2_ is the weight of noodles after cooking (g).

#### 2.3.3. Broken Ratio

The cooked noodles were collected at the optimum cooking time. The broken ratio was calculated as the ratio of the number of broken noodles to the total noodles.

#### 2.3.4. Cooking Loss

The noodles (25 g) were boiled in 500 g of water and then dried to collect solid matter. The cooking loss of noodles was calculated as follows:Cooking loss (%) = *m*_3_/*m*_1_ × 100
where *m*_1_ is the weight of noodles before cooking (g), and *m*_3_ is the weight of solid matter (g).

### 2.4. Texture Profile Analysis (TPA)

TPA was evaluated using a TA-XT2 Texture Analyzer (NR10QC, 3NH Technology Co., Ltd., Shenzhen, China) according to the reported method [13]. Two cooked noodle strands were placed parallel on a flat plate and compressed to 75% of the original height at a speed of 1 mm/s using the probe P/36 R. Measurements were performed with six replicates.

### 2.5. Sensory Evaluation

The noodles were boiled for the optimum cooking time and presented to 30 trained panelists who evaluated their acceptability in terms of color, appearance, smoothness, firmness, elasticity, stickiness, flavor, and overall acceptability according to the reported method [12].

### 2.6. Scanning Electron Microscopy (SEM) Analysis

Uncooked noodles and raw flour were lyophilized in a freeze-drying machine (ALPHA 1-2LD PLUS, CHRIST, Osterode, Germany). The fractured noodles and raw flour were attached to a copper platform with double-sided tape. The fractured surfaces of the noodles were sprayed with gold and were observed using the scanning electron microscope (SU-1510, HITACHI Co., Ltd., Tokyo, Japan).

### 2.7. Water Distribution in Cooked Tartary Buckwheat Noodles

Water distribution in the cooked buckwheat noodles was determined according to our previous method [14]. The transverse relaxation time (T2) of noodles was measured by the Carr-Purcell-Meiboom-Gill (CPMG) sequence using a Low-Field Nuclear Magnetic Resonance analyzer (LF-NMR, MicroMR-25, Niumag Analytical Instruments Co., Ltd., Suzhou, China). Determination parameters were set as follows: scanning frequency (SF) = 22 MHz, number of sampling points (TD) = 25,028, interval time of sampling (TW) = 1500 ms, number of slices (NS) = 32, echo time (TE) = 0.100 ms, and NECH = 1000.

### 2.8. Degree of Starch Gelatinization (DG) and Retrogradation (DR)

DG and DR were determined according to the reported method [15,16] with minor modifications. The noodles for DG determination were mixed with 40 mL 0.05 mol/L KOH, shaken for 20 min, and then centrifuged at 4000× *g* for 10 min. The supernatant was adjusted by 0.05 mol/L HCl for further absorbance determination. As for DR determination, the noodles were mixed with α-amylase in the phosphate buffer (pH 6.0, 0.1 mol/L), followed by incubation at 37 °C for l0 min. The enzymatic reaction was stopped by adding 4 mol/L NaOH. The obtained solution was used for further absorbance determination. Both DG and DR solutions were mixed with iodine solution (0.2% I_2_ − 2% KI) and measured at 625 nm. The DG (%) was calculated as the ratio of the absorbance of noodles to the control, which consisted of KOH (0.5 mol/L) and HCl (0.5 mol/L). The DR (%) was calculated as follows:DR (%) = (*A*_2_ − *A*_3_)/(*A*_1_ − *A*_3_) × 100
where *A*_1_ is the absorbance of the solution of noodles without α-amylase, *A*_2_ is the absorbance of noodles, and *A*_3_ is the absorbance of the solution of noodles after digestion without NaOH.

### 2.9. X-ray Diffraction (XRD)

The crystalline phase of uncooked noodles was analyzed using an X-Ray Polycrystalline Diffractometer (D8 Advance, Bruker Instruments Ltd., Sarbrücken, Germany) using Cu Kα radiation (λ = 1.54056 nm) within the 2θ range of 5° to 50°. The relative crystallinity (RC) of the noodles was calculated by the ratio of the crystalline to amorphous phases using TOPAS (Bruker Instruments Ltd., Sarbrücken, Germany) software.

### 2.10. Fourier Transform Infrared (FTIR) Spectroscopy

The short-range ordered structure of starch was determined using an FTIR spectrometer (IS50, Thermo Fisher Scientific, Inc, Waltham, MA, USA). Freeze-dried noodles (1 mg) and 150 mg of potassium bromide were accurately weighed and ground. The powder was pressed to a uniform and transparent sheet scanned in the wavenumber range of 400–4000 cm^−1^ at a resolution of 4 cm^−1^. The results were expressed as the absorbance ratio of 1047/1022 cm^−1^ and 1022/995 cm^−1^.

### 2.11. Determination of Flavonoids Content and Antioxidant Activities

#### 2.11.1. Total Flavonoid Content

Noodles (2 g) in 20 mL of 70% methanol were treated by ultrasonic extraction for 15 min and put into a shaker at 200 r/min for 1 h. Then, the mixture was centrifuged at 2000× *g* for 12 min to collect the supernatant. The extraction was repeated three times in the dark. The extracts were dried using a vacuum evaporator and stored at 4 °C to further determine total flavonoid contents according to a reported method [15]. The dried flavonoid extracts (1 g) were dissolved into 25 mL of 70% methanol for the content analysis. The total flavonoid content was calculated based on a calibration curve of rutin standards (0–144 µmol/L, y = 6.4782x + 0.0419, R^2^ = 0.994).

#### 2.11.2. Rutin and Quercetin Content

The analysis of rutin and quercetin was based on the reported method [15] with some modifications. The dried flavonoid extract (1 mg) was mixed with 1 mL of methanol and ultrasonicated for 10 min. The contents of rutin and quercetin in the noodles were quantitatively analyzed using High-Performance Liquid Chromatography (HPLC) with a UV detector (1260 Infinity II, Agilent Technologies, Santa Clara, CA, USA) and Waters C18 column (150 mm × 4.6 mm, 5 μm). The injection volume was 20 μL, and the column temperature was 30 °C. The UV detection wavelength was 280 nm. The mobile phase A was 1% acetic acid solution, the mobile phase B was methanol, and the flow rate was 0.6 mL/min. The gradient elution included 90% A for 0–30 min, 65% A for 30–55 min, 50% A for 55–60 min, 90% A for 60–65 min, and 90% A for 65–70 min.

#### 2.11.3. DPPH Radical Scavenging Activity

The DPPH radical scavenging activity was determined by a reported method [17] with slight modifications. The Trolox solution (50 μL) at different concentrations (0–30 mg/mL) and 200 μL 2 mmol/L DPPH solution were mixed and reacted in the dark for 30 min. The absorbance was measured at 517 nm. The standard curve was drawn with the Trolox concentration as abscissa and absorbance value as ordinate. The antioxidant value (mmol TE eq/g DW) of the flavonoid extract from the noodles was expressed as the Trolox equivalent.

#### 2.11.4. ABTS Radical Scavenging Capacity

The ABTS radical scavenging activity assays were conducted according to the reported method [17] with some modifications. The 7 mol/L of ABTS solution and potassium sulfate solution were mixed at a volume ratio of 2:1 and then reacted in the dark for 12 h. The mixture was diluted with anhydrous ethanol until the absorbance value was within 0.70 ± 0.02 at 734 nm. The Trolox solution (20 µL) at different concentrations (0–30 mg/mL) with 500 µL ABTS was reacted in the dark for 10 min. The absorbance was measured at 734 nm. The antioxidant value (mmol TE eq/g DW) of the flavonoid extract was also expressed as the Trolox equivalent.

### 2.12. Statistical Analysis

The data were expressed as means ± standard deviation and analyzed using SPSS 23.0 (IBM Corp., Armonk, NY, USA). Significant differences were determined at *p* < 0.05 by Duncan’s test. Pearson correlation coefficients and Principal Component Analysis (PCA) were analyzed using the correlation procedure and conducted by Origin 2021 (OriginLab Corp., Northampton, NC, USA).

## 3. Results and Discussion

### 3.1. Effect of Frozen Treatment on Cooking Quality of Noodles

The cooking quality of Tartary buckwheat noodles was evaluated in terms of optimum cooking time, water absorption, broken ratio, and cooking loss. The optimum cooking time of TBN was 3 min, while that of both DTBN and FTBN were significantly (*p* < 0.05) prolonged to 17 min and 7 min, respectively (Table 1). However, the optimum cooking time of FTBN was significantly (*p* < 0.05) shorter than that of DTBN, indicating that FTBN was easier to cook than DTBN.

Water absorption, as an important property of starch-based noodles, represents the degree of starch swelling during cooking [18,19]. The water absorption of TBN was 61.56%, while the value of DTBN and FTBN were significantly (*p* < 0.05) increased by 208.30% and 26.87%, respectively. It indicated that both hot air drying and frozen treatments could enhance the water absorption of noodles. Moreover, DTBN showed the highest water absorption of the three groups during cooking due to low original water content, but it might result in the highest broken ratio and cooking loss (*p* < 0.05), contributing to the softness of noodles. The broken ratio and cooking loss of DTBN were increased by 466.60% and 165.86%, respectively. These were closely related to the unfavorable changes in the texture profile and structure of noodles during hot air drying. However, FTBN exhibited no broken ratio, and the cooking loss was not significantly (*p* > 0.05) different from TBN. Therefore, frozen treatment maintained the high quality of fresh extruded Tartary buckwheat noodles. The cooking loss represents the extent of starch leaching or water-soluble substance dissolution during cooking [20]. The long cooking time and excessive water absorption during boiling were not beneficial to the cooking quality of Tartary buckwheat noodles, which was further confirmed by the following texture profiles and sensory evaluation.

### 3.2. Effect of Frozen Treatment on Textural Quality of Noodles

Compared with TBN, the hardness, gumminess, chewiness, and resilience of DTBN were significantly (*p* < 0.05) reduced by 23.53%, 39.55%, 34.56%, and 24.32%, respectively (Table 1). The springiness and cohesiveness of TBN were not significantly (*p* > 0.05) different from DTBN and FTBN. This might be because the interaction between water and starch in noodles increased with the prolonged cooking time. Furthermore, the hardness and chewiness decreased with the prolonged cooking time, since the overhydration destroyed the structure of gelatinized starch and shredded the aggregative starch [3,21]. The overhydration of starch in DTBN with the longest optimum cooking time might be the reason why the gelatinized starch dispersed to the boiling water, which was consistent with the sharp cooking loss and broken ratio in this study. However, different from DTBN, FTBN retained almost the same level of textural quality as TBN. It supported the idea that frozen treatment maintains the cooking quality of fresh noodles by improving the texture profiles.

### 3.3. Different Acceptance in Sensory Quality of Frozen Noodles from Dried Noodles

Sensory evaluation was used to determine the most intuitive consumer acceptance of Tartary buckwheat noodles [22]. As shown in Table 1, the sensory evaluation scores of FTBN and TBN were similar. The color, appearance, and smoothness of the noodles are vital parameters since they are responsible for the first impression of consumers [20,21]. The color scores of the three kinds of Tartary buckwheat noodles were all nearly 7 points, and the color of the noodles was uniform as bright yellow green. The appearance and smoothness of FTBN were the same as that of TBN, while DTBN was much lower, which was directly affected by the expansion of starch gelatinization expansion after water absorption by noodles and the smoothness of their surface.

Firmness, elasticity, and stickiness evaluation were the evaluation of the noodle’s texture. The higher scores of FTBN and TBN ranged from 7 to 9 points and were higher than those of DTBN, around 5–7 points (*p* < 0.05). This showed that the frozen treatment enhanced the texture profiles of the noodles, which presented a chewy but not sticky teeth taste, which, in agreement with the above, resulted in hardness, gumminess, and chewiness.

The flavor scores were 6.40 and 6.55 points of TBN and FTBN, while only 5.20 points of DTBN were probably due to the uniquely bitter taste of Tartary buckwheat, which was generally difficult to be accepted by the public. The overall acceptance of FTBN and TBN was the same at about 8 points, also supporting that the frozen treatment maintained the sensory quality and acceptability of Tartary noodles at the same level as the fresh product.

### 3.4. Change in the Microstructure of Noodles

The microstructures of flour and noodles made from the whole Tatary buckwheat are shown in Figure 1. The starch granules of the whole flour showed smooth surfaces and were tightly packed in clusters (Figure 1A). The starch of extruded noodles was gelatinized. The intact granules disappeared to form the tight structure of starch gels, which might contribute to the suitable hardness and cooking quality of TBN (Figure 1B). Hot air drying and frozen treatment promoted the gaps or pores in the starch gel of noodles (Figure 1C,D). The DTBN shrank to form a more compact and denser structure than FTBN, which generated gaps or cracks in the subsequent starch gels. The massive water loss of starch gels induced by hot air-drying disintegrated starch gel [6]. Moreover, the uneven pores and cracked fragments of DTBN made it difficult for water to enter during boiling, thus prolonging the cooking time. Compared with DTBN, some larger pores and fewer starch fragments were found in FTBN, leading to an increase in the speed of water absorption and a decrease in the dissolution of starch during boiling, namely a shorter cooking time, lower cooking loss, and less broken ratio as mentioned above. Therefore, the significant changes in water migration affected the cooking qualities of the noodles.

### 3.5. Changes in Water Distribution of Cooked Tartary Buckwheat Noodles

The internal microstructure of noodles directly affected the water distribution of noodles during boiling. The transverse relaxation time (T2) was observed, and two different water phases were shown in the present study, including bound water expressed as T21 (0.1–10 ms) and free water as T23 (10–1000 ms) (Figure 2A). In the fresh noodles, a short T2 indicates the close bond between the water and non-water components in noodles, while a long T2 indicates more free water [21]. The proportion of free water was calculated by the integral area of the T23 peak area in TBN (85.23%) and FTBN (84.77%), which were much lower than that of DTBN (86.88%) (*p* < 0.05, Figure 2B). Since the T23 data reflected the water mobility and the interaction between the water molecules and starch gels [14,23], the FTBN probably better embedded the water molecules into the matrix of swelling starch network in the noodles than DTBN in the process of rehydration during boiling. It was reported that frozen treatment could inhibit water loss by forming ice crystals and bound water during storage [24]. It was beneficial for further rehydration, avoiding the disintegration of starch gel during noodle boiling. The water molecules quickly bound to the starch granules in FTBN during rehydration, reducing the cooking time. These results indicated that FTBN could maintain a similar water rehydration to TBN, but DTBN did not. Considering the water absorption results above, the influence of bound water was more important for the cooking quality of Tartary buckwheat noodles than the effect of water content.

### 3.6. Roles of Starch Retrogradation in Cooking Quality of Tartary Buckwheat Noodles

Both starch gelatinization and retrogradation were measured as DG and DR, respectively (Figure 3A). There was no significant difference in DG among the three groups, which was higher than 80% in all (*p* > 0.05). The DG was reported to determine the starch swelling and adhesion properties in the process of extrusion and was closely associated with the water absorption of spaghetti [25]. The same as the spaghetti, the starch in Tartary buckwheat noodles was already gelatinized based on DG results, which means the noodles are edible with appropriate water absorption.

However, the DR level of DTBN was significantly (*p* < 0.05) different from others, although the DG was the same. As the same as the results of texture profiles and water distribution results, the DR of DTBN was the lowest of noodles (*p* < 0.05), but there was no significant (*p* > 0.05) difference between TBN and FTBN. The same level of DR between TBN and FTBN might be ascribed to the fresh noodles being stored at 4 °C, avoiding quality deterioration. The gelatinized starch was the easiest to retrogress at 2~4 °C [24], suggesting that the present study about frozen treatment showed the strongest starch retrogradation. In addition, the frozen time also affected the starch retrogradation. The longer the freezing time was, the greater the starch retrogradation rate was, but when the frozen storage period exceeded 2 day, the increase in the retrogradation rate started to slow down [26]. Therefore, the Tartary buckwheat noodles in this study were stored at −20 °C for 48 h. This process of frozen treatment promoted the starch granule rearrangement and the formation of micelles by directional arrangement. This is the reason why the hardness and tensile properties of FTBN were much stronger than that of DTBN. Furthermore, water is also necessary. The water content being around 40% promoted the starch retrogradation best because there were enough of the interactions of water molecules with starch granules to form the double helix of starch molecule chains [27]. Meanwhile, the extruded buckwheat noodles with 40% water content showed the highest total flavonoid preservation in the range from 30% to 50% water content [4]. This is the reason why a water content of 40% was added in the process of noodle extrusion. FTBN kept water better than DTBN, promoting the better rearrangement of starch molecules. It was illustrated that the starch molecule rearrangement in the process of frozen treatment was necessary for the stability of starch gel and thus played a vital role in the ultimate cooking quality of starch-based Tartary buckwheat noodles.

### 3.7. Effect of the Crystalline Starch Structure of Noodles on Different Treatments

The XRD patterns showed that the main diffraction peaks were at 17° and 20.3° (Figure 3B). Obvious changes were evident in the structure of the Tartary buckwheat starch after extrusion treatment, which was consistent with the crystal structure characteristics of V-type starch [28]. There were no new diffraction peaks that appeared in DTBN and FTBN after hot air drying or frozen treatment. As reported, different drying methods, such as hot air drying and freeze-drying, only changed the peak intensity rather than the crystal type [29]. Compared with TBN, the RC of DTBN significantly (*p* < 0.05) decreased by 42.73%, which was consistent with the results of DR. It might be attributed to the fact that hot air drying interfered with the rearrangement of starch [30]. It was in agreement with the findings of pasta that the formation of an orderly crystal structure of starch was destroyed by heating at 60 °C, and thus the degree of crystallinity decreased during pasta preparation [31]. However, the crystallinity of FTBN was not different from that of TBN (*p* > 0.05), further confirming the improvement of frozen treatment on starch retrogradation and recrystallization [32]. Proper starch retrogradation allowed FTBN to maintain the same quality as TBN.

### 3.8. Changes in the Short-Range Starch Structure of Noodles

The short-range ordering of starch refers to the formation of a double helix structure by the short chains or side chains of amylose and amylopectin in starch molecules, which can be reflected by the intensity ratio of the absorption peaks at 1047 cm^−1^, 1022 cm^−1^, and 955 cm^−1^ in the infrared spectrum [30]. The 1047/1022 cm^−1^ ratio of TBN was 1.0585, and that of DTBN and FTBN decreased significantly (*p* < 0.05) by 1.51% and 0.66%, respectively (Figure 3C,D). Moreover, the 1047/1022 cm^−1^ ratio of FTBN was significantly (*p* < 0.05) higher than that of DTBN, and the 1022/955 cm^−1^ ratio was opposite. It was found that the higher the 1047/1022 cm^−1^ ratio and the lower 1022/955 cm^−1^ ratio corresponding to the short-range ordering of starch, the higher the relative crystallinity and DR [33]. The short-range ordering of starch represents the formation of a double helix structure composed of short chains or side chains of amylose and amylopectin in starch molecules [34]. It explained the findings above, that water maintenance induced by frozen treatment could promote the double helix structure. These results supported that the short-range order structure of FTBN was stronger than that of DTBN, subsequently confirming that frozen treatment improved the cooking quality of the noodles by promoting starch recrystallization and the stability of starch gel.

### 3.9. Effect of Frozen Treatment on the Functional Property of Tartary Buckwheat Noodles

Flavonoids are the most important bioactive compounds in Tartary buckwheat. The total content of flavonoids in TBN was 21.41 mg/g (Figure 4A), which was consistent with the reported raw materials of Tartary buckwheat and its noodles [2,5,15]. As shown in Figure 4B, the peak time of rutin and quercetin in the HPLC chromatogram was 34.128 min and 48.025 min, respectively. The contents of rutin and quercetin were 5.39 mg/g and 14.67 mg/g, respectively (Figure 4C). Both hot air drying and frozen treatment significantly (*p* < 0.05) decreased the contents of total flavonoids, rutin, and quercetin in the noodles. However, the total flavonoid, quercetin, and rutin contents of FTBN were significantly (*p* < 0.05) higher than that of DTBN, indicating that frozen treatment was beneficial to protect the antioxidant nutrients in noodles, which was consistent with the previous results of Tartary buckwheat [15,35]. Moreover, the content of quercetin in all noodles was higher than that of rutin, which might be due to the conversion of rutin to quercetin under the action of rutin during the preparation of noodles. It thus might increase the bitterness [36].

The results of DPPH and ABTS scavenging activity were consistent with total flavonoid, quercetin, and rutin contents. The DPPH and ABTS scavenging activity of TBN was the highest of all noodles (Figure 4D). The DPPH scavenging activity of DTBN and FTBN decreased significantly (*p* < 0.05) by 41.01% and 22.56%, respectively, while a substantial decline (*p* < 0.05) of 58.86% and 23.42% was evident in the ABTS scavenging activity. Moreover, a significant (*p* < 0.05) positive relationship was apparent between the total flavonoid content and antioxidant capacity (DPPH and ABTS). These results further indicated that frozen treatment effectively protected the antioxidant nutrients in the noodles and thus effectively retained the antioxidant capacity.

### 3.10. Correlation Analysis on Noodle Quality with Frozen Treatment and Hot Air Drying

Pearson correlations between noodle quality and water or starch properties are shown in Figure 5A, and PCA was used to visualize the impact of the different treatments on the sensory and functional qualities of noodles (Figure 5B). PC1 (13.2%) and PC2 (70.6%) accounted for 83.6% of the total variance, indicating the reliable main contribution of variables. Free water showed a positive correlation with cooking loss and broken ratio, but bound water was the opposite, which supported the findings that bound water was beneficial for the cooking quality of extruded noodles. Similar to texture profiles and sensory quality, the positive correlation of bound water suggested its key role in the quality improvement of TBN. The interaction between water molecules and starch granules was strengthened, and the increase in the bound water ratio might promote the ordered arrangement of starch and the structural stability of noodles. The important roles of starch retrogradation were also confirmed by the close correlations of DR, RC, and the peak ratio of 1022/955 with cooking, textural and sensory qualities (Figure 5A). The changes in the sensory and functional qualities of TBN and FTBN showed the same positive correlation as PC1, but not DTBN (Figure 5B). According to the above findings, the schematic diagram of the different influences on noodles quality between frozen treatment and hot air drying was created (Figure 6). The maintained ice crystals in the noodles induced by freezing promoted the starch gelatinization evenly during boiling, as well as uniform retrogradation during cooling afterward. The bound water enhanced the stability of the starch gel structure. The water distribution is critical for the starch molecular rearrangement, which resulted in the frozen treatment that protected the noodles from serious cook loss, and they did not like hot air drying. The stable quality of FTBN allowed the better preservation of flavonoid content and antioxidant capacity of noodles, maximizing the health benefits of Tartary buckwheat.

## 4. Conclusions

The treatment of gluten-free noodles after extrusion plays a key role in the cooking and sensory qualities. As the most typical industrial method, hot air drying increases the broken ratio (466.6%) and cooking loss (165.86%) of TBN, but frozen treatment as an alternative technique can protect the noodles from quality deterioration based on the lower broken ratio (only 0%) and cooking loss (5.85%) of FTBN than DTBN, even lower than TBN. The sensory quality of FTBN is much better than DTBN and reaches the same level as TBN. These improvements of FTBN correspond to the same results of texture profiles, in which the hardness, gumminess, chewiness, and resilience are kept the same as TBN. The mechanism of frozen treatment on the noodle texture profiles is closely relevant to its effects on water distribution. The increase in bound water is beneficial to the rapid rehydration of noodles during boiling. On the one hand, water migration within a short cooking time during boiling is crucial for good quality. The SEM morphological analysis indicates that the conversion of ice to water and the formation of small pores increase. On the other hand, the interaction between water molecules and starch granules during starch retrogradation may produce enough bound water, reinforcing the starch gel structure of noodles when boiling frozen noodles. Starch recrystallization and retrogradation are promoted by frozen treatment rather than hot air drying, according to the results of DR, XRD, and FTIR. Although both treatments decreased the flavonoid content and antioxidant capacity of cooked noodles, frozen treatment is preferred to avoid the loss effectively. Therefore, frozen treatment is a feasible way to maintain the quality of extruded Tartary buckwheat noodles by strengthening and stabilizing the structure of the starch gel. Although frozen noodles can meet the sensory requirements of consumers and has shown good economic benefits in the Chinese market, the frozen treatment less than 48 h followed by drying again should be studied in the future to utilize the best promotion of starch retrogradation and become more environmental-friendly.

## Figures and Tables

**Figure 1 foods-11-03989-f001:**
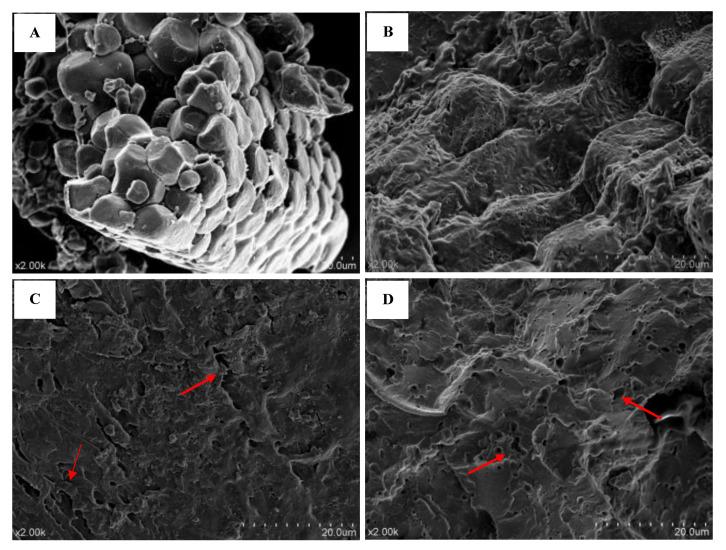
Scanning electron microscopy of whole Tartary buckwheat flour (**A**), fresh Tartary buckwheat noodles (**B**), dried Tartary buckwheat noodles (**C**), and frozen Tartary buckwheat noodles (**D**).

**Figure 2 foods-11-03989-f002:**
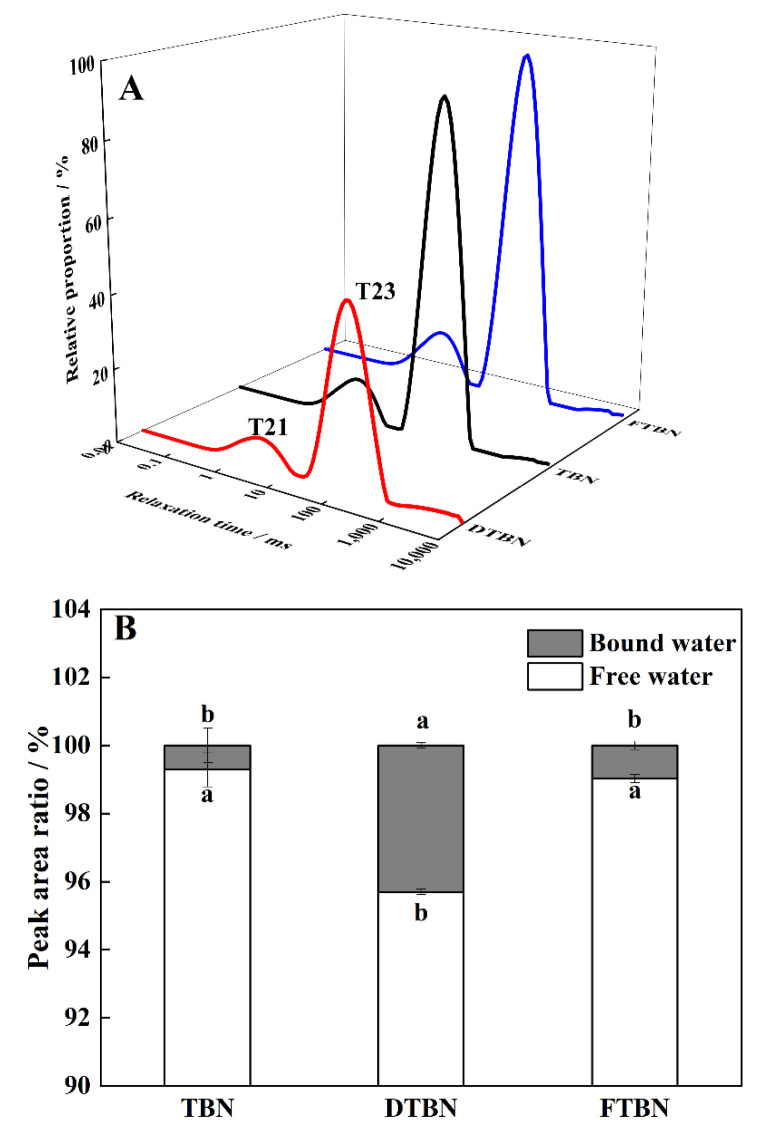
Water distribution of Tartary buckwheat noodles boiled at the optimum cooking time. Curves of T2 relaxation time (**A**); the peak area ratio of T2 (**B**). TBN, fresh Tartary buckwheat noodles; DTBN, dried Tartary buckwheat noodles; FTBN, frozen Tartary buckwheat noodles. Different letters in the same figure imply that the values are significantly different (*p* < 0.05).

**Figure 3 foods-11-03989-f003:**
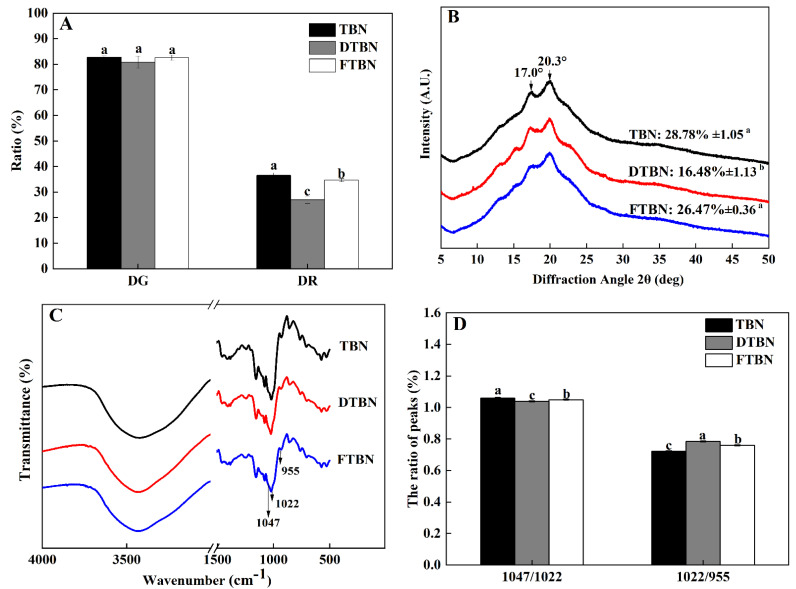
Changes in starch properties and the structure of Tartary buckwheat noodles. Starch gelatinization and retrogradation (**A**); X-ray diffraction (**B**); Fourier transform infrared (**C**); the absorbance ratios at 1047/1022 and 1022/955 cm^−1^ (**D**). DG, degree of starch gelatinization; DR, degree of starch retrogradation; TBN, fresh Tartary buckwheat noodles; DTBN, dried Tartary buckwheat noodles; FTBN, frozen Tartary buckwheat noodles. Different letters in the same figure imply that the values are significantly different (*p* < 0.05).

**Figure 4 foods-11-03989-f004:**
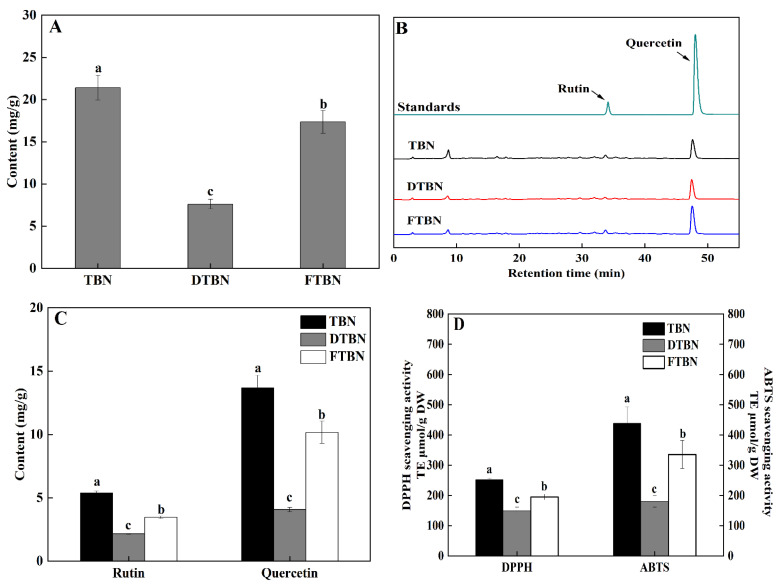
Functional property of Tartary buckwheat noodles. Total flavonoid content (**A**); High-Performance Liquid Chromatography (HPLC) of quercetin and rutin (**B**); quercetin and rutin contents (**C**); DPPH and ABTS scavenging activities (**D**). TBN, fresh Tartary buckwheat noodles; DTBN, dried Tartary buckwheat noodles; FTBN, frozen Tartary buckwheat noodles. Different letters in the same figure imply that the values are significantly different (*p* < 0.05).

**Figure 5 foods-11-03989-f005:**
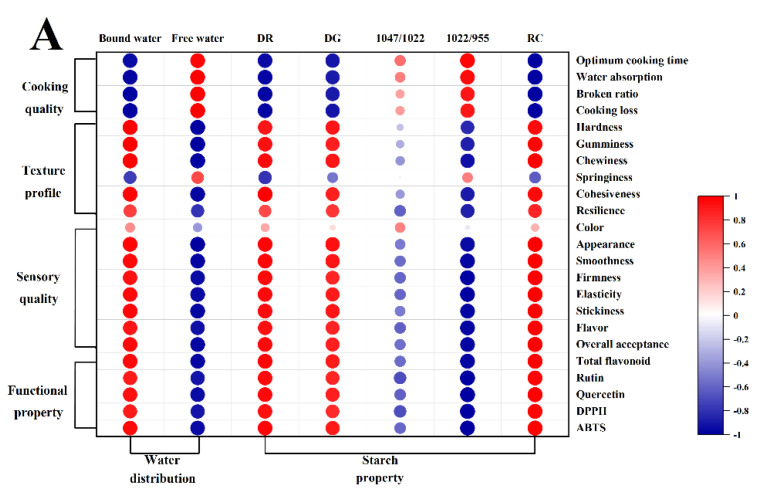
Association of quality of Tartary buckwheat noodles with the changes in the water distribution and starch properties. Pearson correlation coefficients (**A**) and Principal Component Analysis (PCA) (**B**). DG, degree of starch gelatinization; DR, degree of starch retrogradation; RC, relative crystallinity; TBN, fresh Tartary buckwheat noodles; DTBN, dried Tartary buckwheat noodles; FTBN, frozen Tartary buckwheat noodles.

**Figure 6 foods-11-03989-f006:**
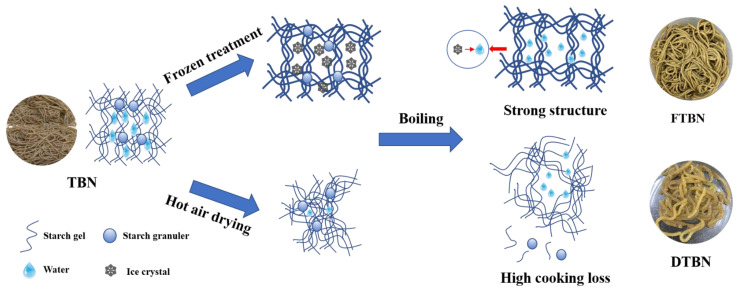
Schematic diagram on the quality of maintenance by frozen treatment and deterioration by hot air drying extruded noodles. TBN, fresh Tartary buckwheat noodles; DTBN, dried Tartary buckwheat noodles; FTBN, frozen Tartary buckwheat noodles.

**Table 1 foods-11-03989-t001:** Cooking, textural, and sensory qualities of cooked noodles.

		TBN	DTBN	FTBN
Cooking quality	Optimum cooking time/min	3.00 ± 0.10 ^c^	17.00 ± 3.20 ^a^	7.00 ± 0.70 ^b^
	Water absorption/%	61.56 ± 1.68 ^c^	189.79 ± 9.34 ^a^	78.10 ± 2.76 ^b^
	Broken ratio/%	5.00 ± 1.00 ^b^	28.33 ± 3.06 ^a^	0 ± 0 ^c^
	Cooking loss/%	8.23 ± 0.02 ^b^	21.88 ± 0.98 ^a^	5.85 ± 0.64 ^b^
Texture profile	Hardness/g	7195.94 ± 219.01 ^a^	5502.98 ± 362.17 ^b^	8656.99 ± 568.10 ^a^
	Gumminess/g	4773.56 ± 445.38 ^a^	2885.53 ± 186.80 ^b^	5640.66 ± 353.75 ^a^
	Chewiness/g	3676.57 ± 195.00 ^a^	2405.87 ± 172.29 ^b^	3841.75 ± 130.29 ^a^
	Springiness/%	0.77 ± 0.03 ^a^	0.82 ± 0.06 ^a^	0.66 ± 0.11 ^a^
	Cohesiveness/%	0.66 ± 0.09 ^a^	0.52 ± 0.01 ^a^	0.66 ± 0.04 ^a^
	Resilience/%	0.37 ± 0.14 ^a^	0.28 ± 0.04 ^b^	0.34 ± 0.03 ^a^
Sensory quality	Color	7.07 ± 0.99 ^a^	7.06 ± 0.98 ^a^	7.80 ± 0.69 ^a^
	Appearance	8.00 ± 0.66 ^a^	5.50 ± 0.34 ^b^	8.20 ± 0.64 ^a^
	Smoothness	8.40 ± 0.51 ^a^	4.50 ± 0.17 ^c^	7.60 ± 0.45 ^a^
	Firmness	8.70 ± 1.42 ^a^	5.55 ± 0.16 ^b^	7.80 ± 0.87 ^a^
	Elasticity	8.85 ± 0.33 ^a^	6.20 ± 0.48 ^b^	8.40 ± 0.69 ^a^
	Stickiness	7.60 ± 0.51 ^a^	5.20 ± 0.42 ^b^	7.00 ± 0.80 ^a^
	Flavor	6.40 ± 0.30 ^a^	5.20 ± 0.43 ^b^	6.55 ± 0.39 ^a^
	Overall acceptance	8.35 ± 0.58 ^a^	5.20 ± 0.63 ^b^	7.90 ± 0.86 ^a^

Means of triplicate determination ± SD with different letters in a column are significantly different (*p* < 0.05). TBN, fresh Tartary buckwheat noodles; DTBN, dried Tartary buckwheat noodles; FTBN, frozen Tartary buckwheat noodles.

## Data Availability

The datasets generated for this study are available on request to the corresponding author.

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
