# Peer review of "Effect of Frozen Treatment on the Sensory and Functional Quality of Extruded Fresh Noodles Made from Whole Tartary Buckwheat"

_foods, 2022, doi:10.3390/foods11243989_

Round 1

Reviewer 1 Report

The manuscript is well-designed and written, the objectives are clear, and the topic is worthy of investigation.The authors should consider analyzing molecular interactions also with statistical analysis. 

Author Response

Responds to Reviewer

Reviewer #1

Thanks for your nice comments and constructive suggestions. We consider seriously and modified in red all over the text.

Reviewers' comments: The manuscript is well-designed and written, the objectives are clear, and the topic is worthy of investigation. The authors should consider analyzing molecular interactions also with statistical analysis.

Reply: Yes, we agree with your good suggestions. Molecular interactions play a key role in the quality improvement by freezing. Therefore, the interaction between starch granules and moisture molecules was emphasized based on Pearson correlation coefficients and principal component analysis (PCA) results. The relevant analysis were added accordingly in the section of Results and Discussion 3.5 “The water molecules quickly bound to the starch granules in FTBN during rehydration, reducing the cooking time” (lines 299-300); 3.6 “water content around 40% promoted the starch retrogradation best because of the enough interactions of water molecule with starch granules to form the double helix of starch molecule chain….FTBN kept water better than DTBN so that promoting starch molecules rearrangement better” (lines 338-345); 3.10 “The interaction between water molecules and starch granules was strengthen, and the increase in bound water ratio might promote the ordered arrangement of starch and the structural stability of noodles” (lines 419-421). 

Reviewer 2 Report

I have reviewed the manuscript entitled “Effect of frozen treatment on the sensory and functional quality of extruded fresh noodles made from whole Tartary buckwheat”. Manuscripts are well written and contributed to current field of research. However, authors have not explained all required parameters properly for majority of the quality parameters. Have a good eye on grammatical errors and vocabulary as it is not up to the journal standard.

Specific comments/suggestions below:

1.      Rewrite the abstract and conclusion part and include numerical data.

2.      Practical application is to be included in the abstract with the significance/potential use of the technology.

3.      Page 1 Line 38: What different process technology?

4.      Page 2 Line 48/49: How and why authors mentioned that hot air drying is the good for drying efficiency and accurate temperature control? Explain with Latest Citation.

5.      Why authors have taken Tartary buckwheat as research materials?

6.      Page 2 Line 77: The amount of water was added 40%; why justify?

7.      For TPA analysis why authors have considered P/36 R probe?

8.      What is the meaning of using the technology FTIR here in your research?

9.      Redraw the figure 5 ?

10.  Rewrite the conclusion part. Add 2-3 lines of environmental economic assessment. 

11.  Cost and Economic analysis must be added for better understandings of the readers.

Author Response

Reviewer #2:

Thank you for your careful and thoughtful comments and constructive suggestions. Those comments are all valuable and very helpful for improving the manuscript, as well as the important direction to our research. We have revised the comments carefully and marked in red in the text according to your suggestions.

Reviewers' comments: I have reviewed the manuscript entitled “Effect of frozen treatment on the sensory and functional quality of extruded fresh noodles made from whole Tartary buckwheat”. Manuscripts are well written and contributed to current field of research. However, authors have not explained all required parameters properly for majority of the quality parameters. Have a good eye on grammatical errors and vocabulary as it is not up to the journal standard.

Reply: Thanks for your critical comments. According to your suggestions, the quality parameters were explained with many modifications as shown in Results and Discussion, and the professional language editing have been performed all over the text as shown in an attached certificate. In addition, the detailed responses to the comments are listed below.

Response to comments:

  1. Rewrite the abstract and conclusion part and include numerical data?

Reply: Sorry for our incomplete numerical data description in the abstract and conclusion. We have rewritten the abstract (lines 11-27) and conclusion accordingly (lines 449-473).

  1. Practical application is to be included in the abstract with the significance/potential use of the technology?

Reply: Thank you for your helpful comments. We do agree with you that practical application is crucial, so the significance/potential us of the technology has been added in the abstracts (lines 25-27), introduction (lines 61-62) and conclusion (lines 468-471). Considering the more and more popular fresh noodles and frozen noodles in present China accounting for approximately 45% of the total noodle production [1], frozen extruded noodles made from the whole grain with the significant advantages of sensory and functional quality over usual dried noodles have powerful potential market even though frozen treatment increase cost to some extent.

  1. Page 1 Line 38: What different process technology?

Reply: Sorry for the unclear expression. We revised the sentence as “However, the lack of gluten protein in Tartary buckwheat asks for extrusion treatment which is different from the traditional calendering technique of wheat noodles to avoid too much cooking loss and breakage” (lines 38-40). The traditional calendering technique is most commonly used to produce wheat noodles based on the gluten protein, which must consist of dough shaping, rolling and slicing. However, this traditional technique cannot produce the gluten-free noodles with satisfy quality. Thus, the different process of extrusion treatment is applied to produce Tartary buckwheat noodles based on starch gelatinization.

  1. Page 2 Line 48/49: How and why authors mentioned that hot air drying is good for drying efficiency and accurate temperature control? Explain with Latest Citation.

Reply: Thank you for the kind comments. We have added some Latest Citations and explained as “Hot air drying is the commonest method during industrial production for its efficiency and accurate temperature control compared with native air drying [2,3], as well as the lower cost and easier operation than other drying methods, such as microwave, infra-red and vacuum [4]” (lines 49-52).

  1. Why authors have taken Tartary buckwheat as research materials?

Reply: Thank you very much for your valuable comments. The first reason is the special health benefit of Tartary buckwheat. It has unique functional properties, with the most contents of rutin and quercetin of all cereal grains. Our previous study also found that flavonoids and dietary fiber in Tartary buckwheat are beneficial to blood sugar control [5-7]. Second, Tartary buckwheat can be extruded to noodles successfully. We found that quinoa and oats etc. are difficult to produce whole-grain noodles by extrusion due to starch types and anti-nutrient factors. We developed the Tartary buckwheat noodles with glycemic index lower than 55 (Patent No. ZL201610552904.4), which is really popular for the diabetic consumers in the market. Third, noodles are the idea intake for maximizing the function of Tartary buckwheat and extrusion is feasible to preserve flavonoids according to the present results. Therefore, Tartary buckwheat has been studying for more than 10 years in our team.

  1. Page 2 Line 77: The amount of water was added 40%; why justify?

Reply: Thank you for the good question. In fact, we optimized the water addition as 30%, 40%, and 50% of the extruded noodles before the study, as well as the extruded temperature as shown in the Fig. 1 below. We added the explanation in the method “According to our previous research, the amount of added water was 40%, and the speed was 15 Hz. The temperature was controlled at 60-70°C, 70-90°C, 90-100°C, and 70-60°C, respectively” (lines 80-82), and in the results and discussion 3.6 “water is also necessary. Water content around 40% promoted the starch retrogradation best because of the enough interactions of water molecule with starch granules to form the double helix of starch molecule chains [8]. Meanwhile, the extruded buckwheat noodles with 40% water content showed the highest total flavonoid preservation in the range from 30% to 50% water content [9]. That is the reason why water content of 40% was added in the process of noodles extrusion. FTBN kept water better than DTBN so that promoting starch molecules rearrangement better” (lines 338-345).

Fig 1. Effect of different extrusion temperature and water addition of extruded Tartary buckwheat whole flour noodles on cooking quality. Water absorption (A); Broken ratio (B); Cooking loss (C). Group 1, zone 1: 50-60°C, zone 2: 60-70°C, zone 3: 70-80°C, zone 4: 60-70°C; Group 2, zone 1: 60-70°C, zone 2: 70-90°C, zone 3: 90-100°C, zone 4: 60-70°C; Group 3, zone 1: 60-80°C, zone 2: 80-100°C, zone 3: 100-120°C, zone 4: 60-70°C.

  1. For TPA analysis why authors have considered P/36 R probe?

Reply: Thank you for a good question. Based on the reported method of extruded gluten-free rice noodles with buckwheat, the P/36 R probe was used in our experiments, because of their similar processing and formation theory by starch gel [10]. In our previous study, the determination by P/36 R probe of noodles from 2 strands to 5 strands showed good repeatability, and the diameter of 36 mm of probe was suitable for the width range of these noodles together [11]. In fact, not only the P/36 R probe has been used to determine noodles widely [10-13], but also the P/LKB probe is recommended in AACC16-50 methods. We don’t have the latter, so the former was chosen in our experiments.

  1. What is the meaning of using the technology FTIR here in your research?

Reply: Your question let us think more about the discussion of FTIR and descript in detail. FTIR is sensitive to the changes in starch molecule chain and double helix structure. Thus, this technology used here aims to clarify the possibility of double helix formation during starch retrogradation, namely short-range ordered structure of starch, when the recrystallization (namely long-range ordered structure) has been confirmed according to the XRD results. As for our study, it is meaningful to illustrate the difference in starch retrogradation between freezing and hot air drying, which modified as “It was found that the higher 1047/1022 cm-1 ratio and the lower 1022/955 cm-1 ratio corresponding to the short-range ordering of starch, the higher relative crystallinity and DR were. The short-range ordering of starch represents the formation of double helix structure composed of short chains or side chains of amylose and amylopectin in starch molecule. It explained the findings above that water maintenance induced by frozen treatment could promoted the double helix structure” (Lines 371-377).

  1. Redraw the figure 5?

Reply: Thank you for your comments. We redraw Figure 5 (line 435).

  1. Rewrite the conclusion part. Add 2-3 lines of environmental, economic assessment?

Reply: Thank you for your comments. We rewrote conclusion and modified according to your suggestion (lines 448-471).

  1. Cost and Economic analysis must be added for better understandings of the readers?

Reply: Thank you for your comments. We added cost and economic analysis in the abstract (lines 25-27) and conclusion accordingly (lines 468-471).

References

[1] Obadi, M.; Zhang, J.; Shi, Y.; Xu, B. Factors affecting frozen cooked noodle quality: A review. Trends in Food Science & Technology. 2021, 109, 662-673.

[2] Xiang, Z.; Ye, F.; Zhou, Y.; Wang, L.; Zhao, G., Performance and mechanism of an innovative humidity-controlled hot-air drying method for concentrated starch gels: A case of sweet potato starch noodles. Food Chemistry. 2018, 269, 193-201.

[3] Bao, H.; Zhou, J.; Yu, J.; Wang, S. Effect of drying methods on properties of potato flour and noodles made with potato flour. Foods. 2021, 10(5), 1115.

[4] Lin, Q.; Ren, A.; Liu, R.; Xing, Y.; Yu, X.; Jiang, H. Flavor properties of Chinese noodles processed by dielectric drying. Frontiers in Nutrition. 2022, 9: 1007997.

[5] Wang, L.; Wang, L.; Wang, T.; Li, Z.; Gao, Y.; Cui, SW.; Qiu, J. Comparison of quercetin and rutin inhibitory influence on Tartary buckwheat starch digestion in vitro and their differences in binding sites with the digestive enzyme. Food Chemistry. 2022, 367, 130762.

[6] Wu, W.; Li, Z.; Qiu, J. Anti-diabetic effects of soluble dietary fiber from tartary buckwheat bran in diabetic mice and their potential mechanisms. Food & Nutrition Research, 2021:65, 4998.

[7] Qiu, J.; Liu, Y.; Yue, Y.; Qin, Y.; Li, Z. Dietary tartary buckwheat intake attenuates insulin resistance and improves lipid profiles in patients with type 2 diabetes: a randomized controlled trial. Nutrition Research, 2016, 36, 1392-1401.

[8] Qiu, S., Punzalan, M. E., Abbaspourrad, A., Padilla-Zakour, O. I. High water content, maltose and sodium dodecyl sulfate were effective in preventing the long-term retrogradation of glutinous rice grains-A comparative study. Food Hydrocolloids, 2020, 98, 105247.

[9] Sun, X.; Yu, C.; Fu, M.; Wu, D.; Gao, C.; Feng, X.; Cheng, W.; Shen, X.; Tang, X. Extruded whole buckwheat noodles: effects of processing variables on the degree of starch gelatinization, changes of nutritional components, cooking characteristics and in vitro starch digestibility. Food & Function. 2019, 10, 6362-6373.

[10] Fu, M.; Sun, X.; Wu, D.; Meng, L.; Feng, X.; Cheng, W.; Gao, C.; Yang, Y.; Shen, X.; Tang, X. Effect of partial substitution of buckwheat on cooking characteristics, nutritional composition, and in vitro starch digestibility of extruded gluten-free rice noodles, LWT - Food Science and Technology, 2020, 126, 109332.

[11] Zou, S.; Wang, L.; Wang, A.; Zhang, Q.; Li, Z.; Qiu, J. Effect of Moisture Distribution Changes Induced by Different Cooking Temperature on Cooking Quality and Texture Properties of Noodles Made from Whole Tartary Buckwheat, Foods, 2021, 10, 2543.

[12] Ma, M., Han, C., Li, M., Song, X., Sun, Q., Zhu, K. Inhibiting effect of low-molecular weight polyols on the physico-chemical and structural deteriorations of gluten protein during storage of fresh noodles, Food chemistry, 2019, 287, 11-19.

[13] Guo, X., Wu, S., Zhu, K. Effect of superheated steam treatment on quality characteristics of whole wheat flour and storage stability of semi-dried whole wheat noodle, Food Chemistry, 2020, 322, 126738.

Reviewer 3 Report

Noodles are like a staple food in south east countries and prepared from various starch products. In this research, it is prepared by using buckwheat. Normally, these noodles are dried to extend their shelf life, while here authors compare fresh, dried and freeze dried methods for extension of shelf life and then check its acceptability.
This research is original and commercially freeze drying is not used for noodles. According to this research freeze drying can be successfully used without affecting its cooking and eating properties which is a new innovation for industrial purposes but it will increase cost many many fold as compared to traditionally used methods.
It adds a preservation method and then its effect on various characteristics of buckwheat noodles.
The methodology was well written. It covers structural, functional and nutritional properties of food. However, if possible, the digestibility of these noodles can also be analyzed.
Results and Conclusion is fine. And References are also fine and up to date.

Author Response

Reviewer #3:
Thanks for your nice comments and constructive suggestions. We consider seriously and modified in red all over the text. 

Reviewers' comments: Noodles are like a staple food in south east countries and prepared from various starch products. In this research, it is prepared by using buckwheat. Normally, these noodles are dried to extend their shelf life, while here authors compare fresh, dried and freeze dried methods for extension of shelf life and then check its acceptability.
This research is original and commercially freeze drying is not used for noodles. According to this research freeze drying can be successfully used without affecting its cooking and eating properties which is a new innovation for industrial purposes but it will increase cost much many fold as compared to traditionally used methods. It adds a preservation method and then its effect on various characteristics of buckwheat noodles.
The methodology was well written. It covers structural, functional and nutritional properties of food. However, if possible, the digestibility of these noodles can also be analyzed. Results and Conclusion is fine. And References are also and up to date.
Reply: Thank you very much for your comments and suggestions. The present study focuses on the effects of frozen treatment on sensory and textural quality of Tartary buckwheat extruded noodles. As you suggested, starch retrogradation must be closely relevant to its digestibility, which is worthy to be illustrated. We will prefer to focus on digestion and rheological properties of Tartary buckwheat starch in our next research. Thank you again.

Round 2

Reviewer 2 Report

All the suggestions are properly incorporated by the Author. In this version, the author has given more importance to technical accuracy, and hence the manuscript may be accepted.